# ECG Semantic Integrator (ESI): A Foundation ECG Model Pretrained with LLM-Enhanced Cardiological Text

**Han Yu**                                                                                 *hy29@rice.edu*
*Department of Electrical and Computer Engineering*
*Rice University*

**Peikun Guo**                                                                            *pg34@rice.edu*
*Department of Computer Science*
*Rice University*

**Akane Sano**                                                                  *akane.sano@rice.edu*
*Department of Electrical and Computer Engineering*
*Rice University*

**Reviewed on OpenReview:** *https://openreview.net/forum?id=giEbq8Khcf*

## Abstract

The utilization of deep learning on electrocardiogram (ECG) analysis has brought the advanced accuracy and efficiency of cardiac healthcare diagnostics. In this work, we address a critical challenge in the field of ECG analysis with deep learning: learning robust representation without large-scale labeled datasets. We propose ECG Semantic Integrator (ESI), a novel multimodal contrastive pretraining framework that jointly learns from ECG signals and associated textual descriptions. ESI employs a dual objective function that comprises a contrastive loss and a captioning loss to develop representations of ECG data. To create a sufficiently large and diverse training dataset, we develop a retrieval-augmented generation (RAG)-based Large Language Model (LLM) pipeline, called Cardio Query Assistant (CQA). This pipeline is designed to generate detailed textual descriptions for ECGs from diverse databases. The generated text includes information about demographics and waveform patterns. This approach enables us to compile a large-scale multimodal dataset with over 660,000 ECG-text pairs for pretraining ESI, which then learns robust and generalizable representations of 12-lead ECG. We validate our approach through various downstream tasks, including arrhythmia detection and ECG-based subject identification. Our experimental results demonstrate substantial improvements over strong baselines in these tasks. These baselines encompass supervised and self-supervised learning methods, as well as prior multimodal pretraining approaches. Our work shows the potential of combining multimodal pretraining to improve the analysis of ECG signals.

The training code and generated waveform descriptions are available at `https://github.com/comp-well-org/ESI`.

## 1 Introduction

The electrocardiogram (ECG), which provides a non-invasive and comprehensive view of the heart's electrical activity, is an important tool in cardiovascular diagnostics and clinical decision-making(Kligfield et al., 2007). For example, ECG has been extensively used in various clinical scenarios, such as diagnosing cardiovascular diseases (Jain et al., 2014), obstructive sleep apnea (Faust et al., 2016), and Parkinson's disease (Haapaniemi et al., 2001), etc. On the other hand, the rapid development of deep learning has triggered general interest in ECG data analysis using data-driven approaches. These deep learning methods, recognized for their ability to learn complex representations, have been proven highly effective in enhancing the accuracy and predictive

capability of ECG analysis (Hannun et al., 2019). Typically, the initial step in utilizing ECG signals involves extracting features from the raw data, either through conventional feature engineering or more recent deep learning backbones, such as 1D convolutional neural network (CNN) Zhu et al. (2020); Baloglu et al. (2019); Jing et al. (2021) and Transformer models Meng et al. (2022); Behinaein et al. (2021); Natarajan et al. (2020); Guan et al. (2021); Yan et al. (2019). However, these supervised methods often require large-scale and high-quality annotated training samples, which are costly to obtain (Mincholé & Rodriguez, 2019).

To reduce the reliance on extensive annotations, researchers have explored self-supervised learning (SSL) techniques for ECG signals (Eldele et al., 2021; Kiyasseh et al., 2021; Yu et al., 2023; Gopal et al., 2021). These methods utilize the unlabeled data when pretraining deep feature extractors. Nevertheless, the SSL strategies, which include tasks such as aligning different signal views or reconstructing masked segments, mainly focus on signals. This means these SSL methods pay attention mainly to the waveform characteristics and neglect the semantic meanings of the signals. Consequently, there is no guarantee that those methods can effectively learn robust representations during the pretraining phase to enhance the ECG analysis in the downstream tasks.

Other studies leverage multimodal learning approaches and incorporate additional modalities, such as descriptive text, into the pretraining process. This approach has shown excellent pretraining performance by enabling a more nuanced and comprehensive understanding of the data (Radford et al., 2021; Jia et al., 2021; Yu et al., 2022b). Motivated by the success of multimodal pretraining on image-text pairs, researchers have developed similar methods for ECGs paired with other modalities including label text (Li et al., 2024), electronic health records (EHR) (Lalam et al., 2023), and clinical reports (Liu et al., 2024). However, acquiring such modalities in large quantities for ECG can also be costly, as ECG analysis would require expertise-depended semantic information compared to general computer vision tasks. Additionally, the variability in terminology and detail across different ECG datasets and sources causes a challenge when combining multiple data sources for a larger scale of pretraining data.

To address these challenges, we introduce a two-step multimodal contrastive pretraining framework to enhance the representations learned from ECG signals. We propose a retrieval-augmented generation (RAG)-based pipeline, Cardio Query Assistant (CQA), to generate standardized and enriched textual descriptions for ECGs. By leveraging the capability of RAG to retrieve relevant information from ECG textbooks, CQA transforms basic ECG conditions into enhanced text descriptions that include patient demographics and specific waveform characteristics. Further, based on the enriched textual descriptions from CQA, we introduce ECG Semantics Integrator (ESI), a contrastive learning framework with a captioning loss inspired by (Yu et al., 2022b). ESI aligns ECG signals with their corresponding text annotations, which aims to pretrain the encoders for an enhanced semantic understanding of ECG content. Our contributions are summarized as follows:

- We introduce a RAG-based ECG description generation pipeline CQA that constructs descriptive textual context for ECG samples using demographic information and diagnostic conditions.

- We develop an ESI framework with both contrastive and captioning capability in pretraining to train an ECG foundation model on approximately 650,000 12-lead ECG signals.

- Compared to strong baselines including the prior SOTA supervised and SSL methods, our evaluation demonstrates promising performances in arrhythmia detection and ECG-based user identification. For instance, we observe an improvement of 1.6% in AUC scores for diagnosing arrhythmia classes, and a 3.5% improvement in identifying subjects when compared to prior SSL methods.

## 2 Related Work

### 2.1 Multimodal Representation Learning

Recent studies have introduced foundation models for integrating image and text, which involve both visual and vision-language pretraining. Vision-language pretraining aims to learn multimodal foundation models with enhanced performance on downstream vision and language tasks. Models such as CLIP (Radford et al.,

2021), ALIGN (Jia et al., 2021), Florence (Yuan et al., 2021), and LiT (Zhai et al., 2022) use dual-encoder architectures where image-text pairs are mapped into a shared embedding space. These models are trained using a contrastive objective that pulls together embeddings of matched image-text pairs while pushing apart those of unmatched pairs. This approach has proven effective in developing robust and transferable representations for both image and text modalities, which significantly improves the performance on various vision-language tasks.

In parallel, other research has explored encoder-decoder architectures that leverage generative objectives. Models such as CoCa (Yu et al., 2022b), SimVLM (Wang et al., 2021), BLIP (Li et al., 2022), BLIP-2 (Li et al., 2023b) and OFA (Wang et al., 2022) integrate both encoding and decoding processes, which enables the model to generate text from visual inputs. These models use language modeling alongside image-text matching and generate semantically meaningful outputs. This line of study also demonstrates a flexible and powerful potential for multimodal learning and demonstrates strong performance across a variety of vision-language benchmarks.

The power of these multimodal pretraining techniques has not been limited to general image and text data. Researchers have adapted these approaches to medical imaging, particularly in radiography paired with clinical reports (Liu et al., 2023; You et al., 2023; Wan et al., 2024). However, despite the progress in image-based applications, applying these multimodal pretraining methods to ECG signal processing remains relatively underexplored. In this study, we extend these advanced techniques to ECG data for more robust and generalizable representations in clinical settings.

## 2.2 ECG Diagnosis with Deep Learning

### 2.2.1 Supervised Methods

Deep learning applications in ECG diagnosis have drawn significant attention (Liu et al., 2021; Pyakillya et al., 2017; Sannino & De Pietro, 2018; Wagner et al., 2020; Śmigiel et al., 2021; Mostafa et al., 2019). For instance, Śmigiel et al. (2021) proposed a CNN model with additional entropy-based features for arrhythmia classification. Their method achieves an AUC score of 0.91 across five classes. Mostafa et al. (2019) conducted a comprehensive review of deep learning applications in ECG analysis for sleep apnea detection. They highlighted the success of models such as CNNs and recurrent neural networks (RNNs), which achieve over 90% accuracy on specialized datasets. Despite the proven effectiveness of these methods, the acquisition of clinical annotations required for these methods is often expensive.

### 2.2.2 Unimodal Representation Learning in ECG

Given the expensive nature of clinical annotations, there has been a growing interest in pretraining methods designed to reduce reliance on labeled ECG sequences (Sarkar & Etemad, 2020; Mehari & Strodthoff, 2022; Oh et al., 2022). For example, Mehari & Strodthoff (2022) applied well-known SSL frameworks such as SimCLR (Chen et al., 2020), BYOL (Grill et al., 2020), and CPC (Oord et al., 2018) to pretrain models on 12-lead ECG data. These models showed enhanced robustness, reflected in a 2% increase in AUC score for 5-class arrhythmia classification compared to purely supervised models. However, even though pretraining strategies generally provide insights into performance improvement and decrease reliance on labeled data, these methods are often limited by their focus solely on signal waveforms. The emphasis on waveform alone does not ensure the capture of clinically relevant semantic information. As a result, a multimodal approach incorporating both ECG waveforms and corresponding clinical text helps acquire more meaningful and transferable ECG representations for various downstream tasks.

### 2.2.3 Multimodal Representation Learning in ECG

Although few, some studies have begun to explore the alignment of ECG signals with other modalities such as textual descriptions, EHR, and clinical notes (Li et al., 2023a; Lalam et al., 2023; Liu et al., 2024). For instance, Lalam et al. (2023) utilized identical encoders to extract and contrastively align embeddings from ECG, EHR, and clinical notes. The pretrained model showed promising results in clinical diagnosis. Liu et al. (2024) adopted a similar approach to couple ECG signals with clinical notes and enhanced the ECG encoder's

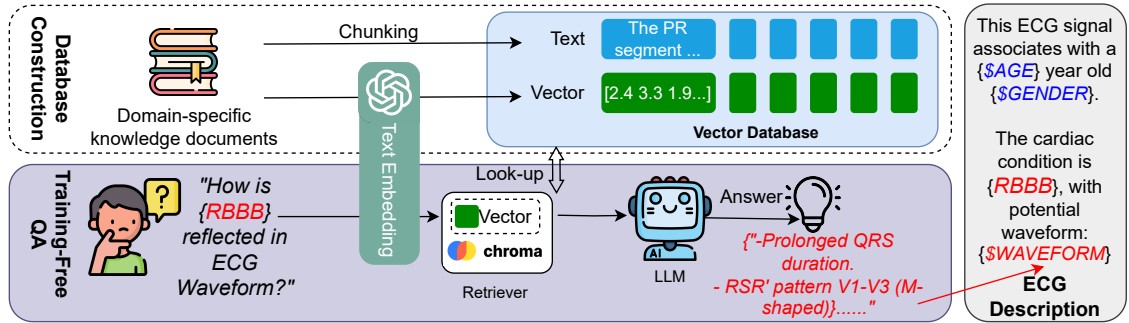

Figure 1: The Cardio Query Assistant (CQA) Framework employs a novel knowledge-based approach to generate detailed and clinically relevant textual descriptions for ECG signals, which translates ECG conditions into enriched ECG waveform patterns.

effectiveness in zero-shot arrhythmia detection. However, the pretraining processes of these methods depend on costly annotations such as clinical notes and EHR, which are challenging to acquire on a large scale. Moreover, the variability in textual descriptions or reports associated with ECGs due to differences in detail, terminology, and style among clinicians and clinical contexts may complicate the learning of consistent mappings between ECG signals and text. The consequent variability could lead to misalignments between the ECG-text pairs. In this study, we propose to utilize a retrieval-augmented generation (RAG)-based pipeline to construct contextual ECG textual data without relying on costly notes and EHRs. Additionally, we introduce a captioning task in our model to achieve more nuanced representations.

## 3 Methods

In this section, we introduce our approach in three main components: (1) RAG-based ECG description pipeline, Cardio Query Assistant (CQA) and (2) the contrastive captioning pretraining framework, ECG Semantics Integrator (ESI).

### 3.1 Cardio Query Assistant (CQA) Framework

The Cardio Query Assistant (CQA) Framework, as shown in Figure 1, is designed to transform ECG condition labels into detailed descriptive text. The generated text incorporates demographic information, ECG conditions, and enriched waveform details. An example of a raw ECG signal, its associated metadata, and the generated ECG waveform description is provided in Figure 2. The developed CQA is outlined as follows:

#### 3.1.1 Establishing a Domain-Specific Knowledge Database

To leverage the enhanced interpretation of ECG conditions with domain expertise, we develop a comprehensive vector database from domain-specific literature of authoritative medical texts guided by two textbooks: (1) *ECG Workout: Exercises In Arrhythmia Interpretation* by Huff (2006), and (2) *12-Lead ECG: The Art of Interpretation* by Garcia (2015). To extract and encode this information into a usable format, we employ the *text-embedding-ada-002* API (OpenAI, 2023) because of its efficiency and performance. The resulting embeddings are then systematically organized using the *Chroma* database management tool, which was chosen for its robustness and ease of integration with the *LangChain* Python library (Mendable, 2023).

#### 3.1.2 Enhancement of ECG Semantics

The CQA Framework enriches the ECG-associated information through a comprehensive retrieval-augmented process. With the pre-constructed domain-knowledge database, the RAG-based approach enables CQA to query related knowledge using given information such as standard clinical labels, standard communications protocol for computer-assisted ECG (SCP) statements, diagnostic interpretations, and machine-generated

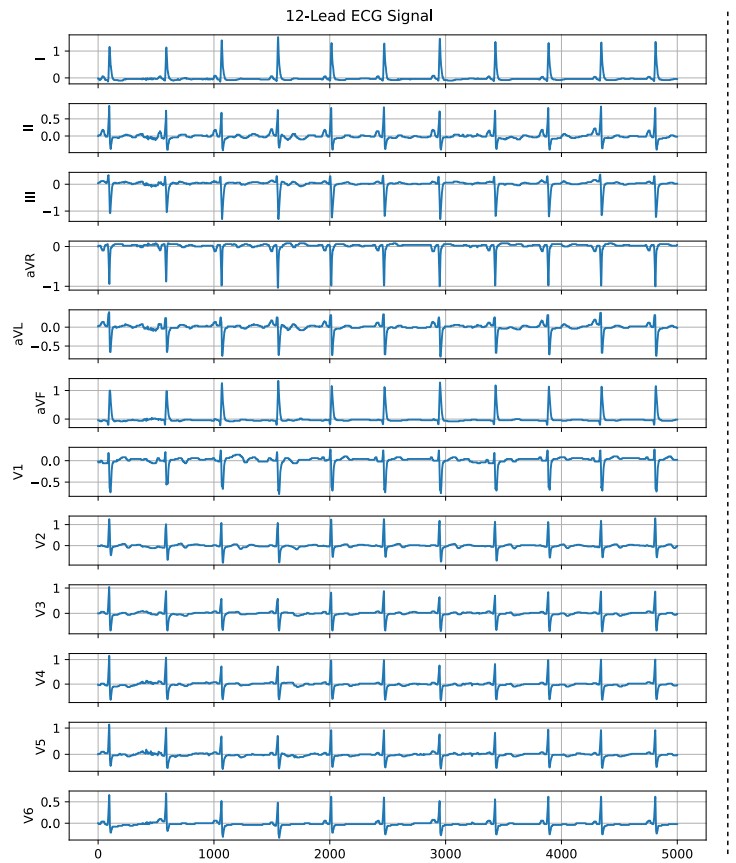

| Patient ID | Age | Gender |
|---|---|---|
| xxx | 47 | M |

ECG Metadata

*Sinus rhythm, Abnormal R-wave progression, early transition, Left ventricular hypertrophy, Nonspecific T abnormalities, diffuse leads, Prolonged QT interval*

ECG Conditions

This patient is a 47-year-old Male.

ECG Findings:
Sinus rhythm, Abnormal R-wave progression, early transition, Left ventricular hypertrophy, Nonspecific T abnormalities, diffuse leads, Prolonged QT interval.

Potential ECG waveforms:
Normal sinus rhythm has a regular rhythm with a rate of 60 to 100 beats per minute. The P waves are normal in size, shape, and direction, positive in lead II. One P wave precedes each QRS complex. The PR interval is normal, between 0.12 to 0.20 seconds. The QRS complex is normal, 0.10 second or less in duration.

In the condition of Abnormal R-wave progression with early transition, the R wave becomes smaller or diminishes across the precordial leads from V1 to V6. The R wave is initially tall in V1 but progressively decreases in amplitude across the precordial leads, becoming smaller than the S wave earlier than expected, typically by V3 or V4............

ECG Metadata

Figure 2: Example of a 12-lead ECG signal and its associated metadata. The left side displays the 12-lead ECG waveform, which provides a visual representation of the heart's electrical activity. The right side includes relevant patient demographic information (age, gender), a summary of ECG metadata including high-level clinical findings (e.g., Sinus rhythm, Abnormal R-wave progression), and a detailed textual description of the ECG conditions generated with the CQA pipeline.

reports associated with ECG data. SCP statements are standardized textual formats that provide consistent documentation of ECG findings, following international protocols for computer-assisted ECG interpretation. While these datasets provide valuable diagnostic information, they often lack explicit details about the waveform patterns that are critical for a thorough ECG analysis.

To address this gap, the CQA Framework employs a RAG approach that enables it to query the knowledge database, which is introduced in the prior section, for relevant information and generate comprehensive textual descriptions of the potential waveform characteristics. For instance, SCP statements and standard arrhythmia diagnoses may not directly include detailed waveform descriptions; however, by querying the domain-knowledge database, the CQA Framework can synthesize this enriched information, producing detailed descriptions of the waveform patterns that correspond to specific ECG conditions.

With the constructed RAG pipeline in CQA, we asked the waveform information by using simple prompts of "How is ECG Condition reflected in a 12-lead ECG"? The output of the CQA framework is organized utilizing an LLM, e.g., GPT-3.5, to generate the potential waveform details from the queried knowledge base. Take an example of the cardiac condition of the Right Bundle Branch Block (RBBB). By executing targeted queries in our database, it retrieves and generates descriptive context for specific waveform attributes. For example, the ECG-associated information of "RBBB" is queried and converted into related waveform features, including "prolonged QRS duration" and "M-shaped RSR' pattern in leads V1-V3."

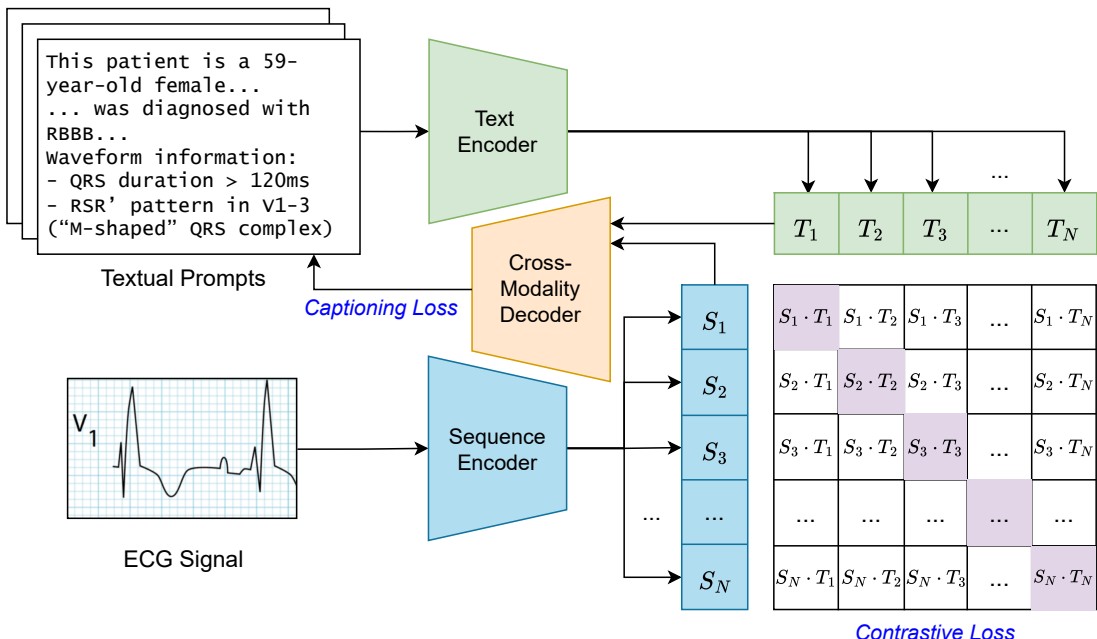

Figure 3: The ECG Semantics Integrator (ESI) is built based on an ECG signal encoder with a text encoder using captioning and contrastive losses for unified representations. This architecture learns from the alignments between detailed textual prompts and the corresponding ECG waveform data, which aims to capture nuanced clinical insights for enhanced diagnostic tasks.

## 3.2 Multimodal Contrastive Captioning with ECG Semantics Integrator (ESI) Framework

The ESI framework aims to improve the quality of representation extracted from ECG signals by pretraining a specialized ECG encoder alongside a textual encoder. This dual-modality training method has been proven by cutting-edge studies in contrastive language-image pretraining (CLIP) (Radford et al., 2021) and CoCa (Yu et al., 2022b) methodology. Inspired by these two studies, we developed our pretraining architecture with both contrastive and generative objectives. While we recognize there are newer approaches in multimodal contrastive learning beyond CLIP and CoCa as discussed in Section 2.1, the choice of development can be motivated by several factors. First, the CLIP structure is well-proven by multiple prior works, including pretraining language with medical images (Liu et al., 2023; You et al., 2023; Wan et al., 2024) and physiological signals (Li et al., 2023a; Lalam et al., 2023; Liu et al., 2024). The core contrastive objective of CLIP is well-suited for our primary goal of aligning ECG signals with their textual interpretations. Moreover, the relative simplicity of the CLIP architecture allows for seamless integration of our captioning loss inspired by CoCa (Yu et al., 2022b). We further validate the contribution of the contrastive and generative objective via ablation study as shown in Section 5.1.

For the ECG encoder, we have chosen a one-dimensional modified version of the ConvNext v2 architecture considering the sequential nature of ECG waveform data (Woo et al., 2023), which has been proven for its capacities of extracting both local and global contexts with the designed convolutional kernels. In parallel, the textual encoder utilizes BioLinkBERT, a derivative of the BERT architecture pretrained on biomedical texts, to effectively embed medical terminologies (Yasunaga et al., 2022). BioLinkBERT is an extension of the standard BERT model (Devlin et al., 2018), specifically designed to improve the understanding of biomedical texts. Unlike traditional BERT, which processes each document independently, BioLinkBERT is pretrained on biomedical literature from PubMed, which takes advantage of the natural links between documents such as citations and references. This pretraining strategy motivates us to leverage BioLinkBert for tasks that require a deep understanding of biomedical concepts and terminology.

### 3.2.1 Multimodal Contrastive Learning

Inspired by the previous vision language pretraining approaches (Radford et al., 2021; Yu et al., 2022b), our framework uses two pretraining objectives for comprehensive learning, including contrastive loss for robust representation learning and captioning loss for semantic alignment.

**Contrastive Loss:** We employ the dual-encoder contrastive learning framework following the prior studies. Compared to pretraining with single-encoder as signal-focused frameworks, e.g., SimCLR (Chen et al., 2020) and BYOL (Grill et al., 2020), the dual-encoder approach in this study leverages the semantic information from the textual modality. Both encoders aim to project the inputting ECG and text into a unified embedding space. Consequently, the two encoders are jointly optimized by contrasting the paired text against others in the sampled batch:

$$\mathcal{L}_{\text{Con}} = -\frac{1}{N}(\underbrace{\sum_{i}^{N} \log \frac{\exp\left(S_i^\top T_i/\sigma\right)}{\sum_{j=1}^{N} \exp\left(S_i^\top T_j/\sigma\right)}}_{\text{ecg-to-text}} + \underbrace{\sum_{i}^{N} \log \frac{\exp\left(T_i^\top S_i/\sigma\right)}{\sum_{j=1}^{N} \exp\left(T_i^\top S_j/\sigma\right)}}_{\text{text-to-ecg}}),$$

with $S_i$ and $T_i$ representing the normalized embeddings from the ECG signal and text encoders for the $i$-th ECG-text pair, $N$ is the batch size during training, and $\sigma$ as the temperature scaling factor. This dual-encoder approach has been working promisingly on enabling cross-modal alignment applications such as zero-shot classification (Radford et al., 2021; Yu et al., 2022b).

**Captioning Loss:** While the dual-encoder approach encodes the text as an embedding for the contrastive learning purpose, the generative approach aims for detailed granularity and requires the model to predict the exact tokenized texts with ECG and preceding texts. This approach encourages the encoders to capture the semantic information embedded in the texts actively. Inspired by the image-text multimodal pretraining study CoCa (Yu et al., 2022b), we design to align the generated textual descriptions with the corresponding ECG signals by additionally defining a captioning loss $\mathcal{L}_{cap}$ similar to that used in image captioning tasks (Vinyals et al., 2015):

$$\mathcal{L}_{Cap} = -\sum_{i}^{N} \log P(t_i|t_{<i}, S_i; \theta),$$

where $t_i$ represents the $i$-th token in the textual description, $t_{<i}$ denotes all the preceding tokens, $S_i$ is the ECG signal, and $\theta$ represents the parameters of both encoders and the cross-modality decoder.

The overall pretraining objective is the combination of both contrastive loss and captioning loss, denoted as:

$$\mathcal{L} = \lambda_{Con} \cdot \mathcal{L}_{\text{Con}} + \lambda_{Cap} \cdot \mathcal{L}_{\text{Cap}}$$

where $\lambda_{Con}$ and $\lambda_{Cap}$ are the loss weighting hyperparameters for the introduced objectives. We set these two weighting parameters equally to 1 in this study. By jointly optimizing these losses, the ESI Framework aims to learn a multimodal representation that enriches the semantic link between ECG waveforms and their textual explanations. This method is anticipated to improve performances in downstream tasks that leverage the waveform details and demographics, such as diagnosing arrhythmia and performing large-scale patient identification using ECG data.

## 4 Evaluation

In this section, we describe our evaluation settings and experimental results. We first introduce the information on the datasets used and tasks performed in this study, along with the baseline methods we used in the comparisons. Our experiments explore three settings: zero-shot learning, linear probing (frozen features), and fine-tuning. In our experiments, we conduct multiple resampling runs to assess the robustness of the model, particularly for the linear probing and fine-tuning settings. Given that the encoder is frozen during linear probing evaluations, the randomness is only applied to the output linear heads during the linear probing setup. For each setting, we average the results over five different runs and report the standard deviations alongside the mean performance metrics.

Table 1: Summary of datasets used in the pretraining stage

| Dataset | Duration | Sampling Rate | # of Training Samples | Associated Information |
|---|---|---|---|---|
| PTB-XL | 10 seconds | 500 Hz | 17K | Demographics, SCP Code |
| Chapman-Shaoxing | 10 seconds | 500 Hz | 45K | Demographics, Arrhythmia Diagnosis |
| MIMIC-IV-ECG | 10 seconds | 500 Hz | 600K | Demographics, Machine-generated ECG Reports |

### 4.1 Training Setup

**Pretraining Datasets**. The proposed ESI signal encoder is pretrained from scratch. Therefore, the pretraining dataset directly impacts the model's generalizability. We constructed a large pretraining set combining three large-scale datasets with over 650,000 ECG-text training pairs. These datasets covers Chapman-Shaoxing (Zheng et al., 2020), PTB-XL (Wagner et al., 2020), and MIMIC-ECG (Gow et al., 2023). Each dataset contains 12-lead and 10-second ECG recordings sampled at 500 Hz. Here is a detailed breakdown of each dataset:

- PTB-XL: This dataset consists of 21,837 12-lead, 10-second ECG recordings from 18,885 participants. We followed the training and test data split guidelines outlined in the original publication (Wagner et al., 2020) and only used the training samples (17k) in the pretraining task. These samples include demographic data and SCP codes.

- Chapman-Shaoxing: This dataset offers a larger set of 45k samples with associated demographic information and arrhythmia diagnoses.

- MIMIC-IV-ECG: This is the most extensive dataset with 600k samples accompanied by demographics and machine-generated ECG reports.

The variety and volume of data provide a comprehensive foundation for the pretraining of models. Table 1 summarizes the overview information of each dataset used in pretraining.

**Implementation**. During the pretraining phase of the ESI model, we make specific choices regarding the encoder architectures, optimizer, learning rate scheduler, training hardware, and batch size. The ECG signal encoder within ESI utilizes a 1D ConvNeXt-base (Woo et al., 2023) backbone as the default architecture. This choice allows the model to effectively capture the spatial features within the ECG signal data. For text encoding, we leverage BioLinkBert (Yasunaga et al., 2022) as the default due to its proven capabilities in handling biomedical text data. The AdamW optimizer is employed for optimization during the pretraining process. We opted for an initial learning rate of $5 \times 10^{-5}$ to facilitate efficient convergence. To further adjust the learning rate throughout training, a warm-up phase of 5 epochs out of the total 30 epochs is implemented. This warm-up phase allows the model to gradually adjust to the training data before applying the main learning rate. Additionally, a learning rate decay of 0.1 is introduced after every 10 epochs to prevent overfitting in the later stages of training.

The pretraining process is conducted on a server equipped with 4 Nvidia A100 GPUs. This hardware configuration provides the computational resources necessary to handle the large datasets used for pretraining efficiently. To leverage the capabilities of these GPUs effectively, a batch size of 48 samples is used on each GPU during training.

In addition to the main ESI model, we implement a parameter-efficient variant named ESI-tiny. This variant utilizes a ConvNeXt-tiny architecture as the ECG encoder, which pretrains a model with a smaller overall size. This can be beneficial in scenarios where computational resources are limited.

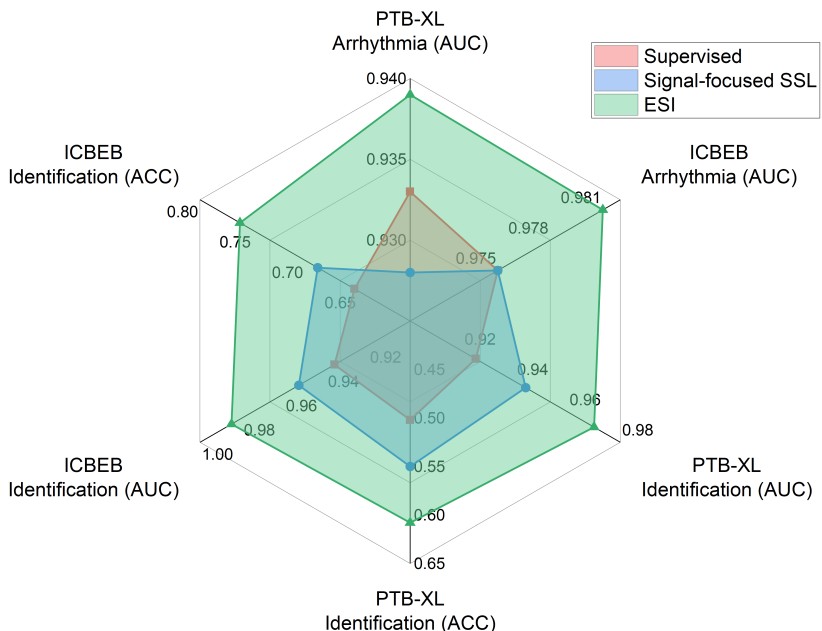

Figure 4: Comparison of the proposed ECG Semantics Integrator (ESI) with the best performances from baseline methods including the supervised models and signal-focused self-supervised learning (SSL) pre-trained models. Compared to the baselines, ESI is a multimodal contrastive pretraining framework that leverages both ECG signals and corresponding textual descriptions to learn enhanced ECG representations. The evaluations of arrhythmia diagnosis and identification are conducted on datasets including PTB-XL and ICBEB, with metrics of area under the ROC curve (AUC) and accuracy (ACC).

## 4.2 ECG Semantics Integrator (ESI) For Downstream Tasks

After the pretraining stage, the encoder can be applied in three different manners, including zero-shot inference, linear probing, and fine-tuning on various downstream tasks. The aim is to validate the robustness of the learned representations and the practical utility of the model in real-world clinical settings. While the zero-shot inference and linear probing can directly assess the representations of the learned framework, the fine-tuned models usually provide the best performances among these three methods, with updating parameters through downstream tasks. As shown in Figure 4, the fine-tuned encoder from our ESI method outperforms the supervised and self-supervised baselines for different downstream tasks.

### 4.2.1 Zero-Shot Evaluation

Zero-shot evaluation assesses the model's capacity to understand and infer information from ECG signals without any task-specific fine-tuning. Following the definition used in the context of CoCa (Yu et al., 2022b) and CLIP (Radford et al., 2021), our zero-shot evaluation strategy ensures that while the model has been exposed to a vast array of ECG and text pairs during pretraining, it has not seen any supervised examples from the downstream tasks. In the ESI Framework, each ECG signal's embedding is compared against a range of possible textual labels for different cardiac conditions without task-specific fine-tuning. The model selects the textual description that has the closest embedding distance to the ECG signal's embedding, which is determined by a similarity metric of cosine similarity. This process demonstrates the model's understanding of ECG data and its ability to correlate it with accurate clinical descriptions directly after pretraining, which aims to demonstrate the potential generalizability of the learned representations without fine-tuning the specific tasks.

### 4.2.2 Linear Probing

Linear probing is a strategy that makes use of the representations learned by the ESI Framework's encoders. In this setup, a linear classifier is utilized on top of the frozen encoders for different downstream tasks such as arrhythmia detection and patient identification. As the only trainable part of the model is the classifier head, thus, the quality and robustness of the representations from the pretrained encoder play an essential role in the linear probing strategy.

### 4.2.3 Fine-Tuning

To introduce more flexibility into the pretrained signal encoder, we can also fine-tune the entire framework on a set of downstream tasks. Similar to the linear probing strategy but with trainable encoders, this fine-tuning strategy aims to explore the full effectiveness of the structure with pretrained parameters as its initialization for downstream tasks.

## 4.3 Downstream Task: Arrhythmia Diagnosis

Cardiac arrhythmias are a significant contributor to cardiovascular diseases, and there is a demand for accurate and reliable detection methods for clinical use. We evaluated our proposed method on arrhythmia detection using two datasets, PTB-XL (Wagner et al., 2020) and ICBEB (Liu et al., 2018). As described in Section 4.1, we follow the training and test data split guidelines outlined in the original PTB-XL publication (Wagner et al., 2020) to divide the PTB-XL dataset. The training set is used for fine-tuning the model and the test set, which is not seen during the pretraining, is used for evaluation. The ICBEB dataset, which is not used during the pretraining, consists of 9,831 12-lead ECG signals from 9,458 patients (Liu et al., 2018). We adopt the processing settings from a prior benchmark study (Strodthoff et al., 2020), which results in 6,877 training samples and 2,954 test samples. Based on this configuration, we evaluate the model's effectiveness across three settings, including fine-tuning, linear probing, and zero-shot learning.

### 4.3.1 Fine-tuning & Linear Probing

To perform a comprehensive evaluation, we compare our proposed method with specialized supervised methods, including a long short-term memory (LSTM), XResNet101, ResNet50, ensemble methods implemented in an ECG benchmark study (Strodthoff et al., 2020), a multi-lead-branch fusion network (MLBF-Net) (Zhang et al., 2021), and a multi-view multi-scale neural network (MVMSN) (Yang et al., 2023). Besides the supervised learning methods, we also cover the comparison between our method and signal-focused SSL methods including SimCLR (Chen et al., 2020), BYOL (Grill et al., 2020), CLOCS (Kiyasseh et al., 2021), and LEAVES (Yu et al., 2022a). The deep learning backbone used for training these methods is ConvNeXt-base, the same as the proposed ESI, to ensure fair comparisons. We perform both the linear probing and fine-tuning strategies for those pretrained methods to assess the quality of learned representations during the pretraining phase. Additionally, we benchmark against MERL (Liu et al., 2024) under a frozen encoder setting as presented in their study.

Table 2 shows the performances of the evaluated methods. In the supervised learning methods, the ConvNeXt and ConvNeXt-tiny encoders show lower performance compared to specialized methods such as MLBF-Net and MVMSN. ConvNeXt-tiny outperforms the larger ConvNeXt model, potentially due to overfitting with the smaller dataset. After pretraining, the ConvNeXt-based ESI method achieves the best performance on both PTB-XL and ICBEB datasets under both the frozen encoder and fine-tuning settings. Notably, the multimodal pretraining methods (ESI and MERL) significantly outperformed the signal-focused methods such as SimCLR and BYOL. This supports our hypothesis that signal-focused approaches may have limitations in learning robust and transferable representations for downstream tasks compared to the multimodal pretraining methods. The superior performance of the pretrained ESI encoder compared to the frozen encoder in supervised learning approaches also demonstrates the robustness of the learned features for arrhythmia diagnosis.

Table 2: Evaluation results of arrhythmia diagnosis task under different settings including supervised learning, linear probing (frozen encoder), and fine-tuning. The metric used is the area under the ROC curve (AUC). The best results are highlighted in **bold**.

| Methods | PTB-XL ($AUC \uparrow$) | ICBEB ($AUC \uparrow$) |
|---|---|---|
| *(Supervised)* | | |
| LSTM (Strodthoff et al., 2020) | $0.927_{\pm 0.013}$ | $0.964_{\pm 0.015}$ |
| XResNet101 (Strodthoff et al., 2020) | $0.928_{\pm 0.013}$ | $0.974_{\pm 0.013}$ |
| ResNet50 (Strodthoff et al., 2020) | $0.930_{\pm 0.015}$ | $0.969_{\pm 0.015}$ |
| Ensemble (Strodthoff et al., 2020) | $0.934_{\pm 0.013}$ | $0.975_{\pm 0.013}$ |
| MLBF-Net (Zhang et al., 2021) | $0.931_{\pm 0.021}$ | - |
| MVMSN (Yang et al., 2023) | $0.933_{\pm -}$ | - |
| ConvNeXt-Tiny (Woo et al., 2023) | $0.918_{\pm 0.016}$ | $0.970_{\pm 0.012}$ |
| ConvNeXt-Base (Woo et al., 2023) | $0.914_{\pm 0.014}$ | $0.970_{\pm 0.012}$ |
| *(Linear Probing)* | | |
| SimCLR (Chen et al., 2020) | $0.766_{\pm 0.016}$ | $0.791_{\pm 0.012}$ |
| BYOL (Grill et al., 2020) | $0.776_{\pm 0.014}$ | $0.800_{\pm 0.014}$ |
| CLOCS (Kiyasseh et al., 2021) | $0.777_{\pm 0.012}$ | $0.802_{\pm 0.015}$ |
| LEAVES (Yu et al., 2022a) | $0.792_{\pm 0.015}$ | $0.809_{\pm 0.013}$ |
| MERL (Liu et al., 2024) | $0.887_{\pm -}$ | - |
| *(Ours)* ESI-tiny | $0.927_{\pm 0.009}$ | $0.975_{\pm 0.006}$ |
| *(Ours)* ESI | $0.931_{\pm 0.008}$ | $0.978_{\pm 0.006}$ |
| *(Fine-tune)* | | |
| SimCLR (Chen et al., 2020) | $0.916_{\pm 0.015}$ | $0.968_{\pm 0.016}$ |
| BYOL (Grill et al., 2020) | $0.925_{\pm 0.014}$ | $0.971_{\pm 0.014}$ |
| CLOCS (Kiyasseh et al., 2021) | $0.918_{\pm 0.013}$ | $0.977_{\pm 0.012}$ |
| CRT (Zhang et al., 2022) | $0.892_{\pm -}$ | - |
| LEAVES (Yu et al., 2022a) | $0.926_{\pm 0.012}$ | $0.976_{\pm 0.013}$ |
| *(Ours)* ESI-tiny | $0.935_{\pm 0.011}$ | $0.978_{\pm 0.010}$ |
| *(Ours)* ESI | $\mathbf{0.938}_{\pm 0.011}$ | $\mathbf{0.982}_{\pm 0.009}$ |

### 4.3.2 Zero-shot Inference

To further assess the learned representations during pretraining, we also evaluate the zero-shot learning inference assessment following METS (Li et al., 2023a) and MERL (Liu et al., 2024). Other than the zero-shot evaluations, we include the few-shot setting of the existing signal-focused SSL approaches in the comparison, including SimCLR (Chen et al., 2020), BYOL (Grill et al., 2020), CLOCS (Kiyasseh et al., 2021), and LEAVES (Yu et al., 2022a), with 5% of the original training set as training samples on PTB-XL dataset.

Table 3 demonstrates the zero-shot learning inference performance of the proposed ESI method alongside baselines (METS, MERL) and the few-shot fine-tuning results for signal-centered SSL methods. Our proposed method achieves the best performance on both AUC and macro F1 scores compared to all other methods. Additionally, ECG-text pretrained models generally outperformed signal-focused pretraining methods even without samples in fine-tuning. This highlights the improved robustness of representations from multimodal pretraining techniques.

### 4.4 Downstream Task: ECG-based User Identification

ECG generally shows unique patterns that can distinguish individuals, which makes them suitable for subject identification tasks and offers potential advantages over other biometric traits (Melzi et al., 2023). For example, compared to identifying persons with facial images, using ECG can further protect users' privacy during usage. In this study, we leverage the PTB-XL (Wagner et al., 2020) and ICBEB (Liu et al., 2018) datasets, also used in the arrhythmia diagnosis task, to design a one-shot learning benchmark for ECG

Table 3: Evaluation results of arrhythmia diagnosis task under different settings in zero-shot learning on PTB-XL dataset. The metric used in this table is the area under the ROC curve (AUC) and macro F1 score (F1-macro). $X - \%$ represents the percentage $X$ of the training set used as the training samples in fine-tuning the pretrained encoder. The best results are highlighted in **bold**.

| Method | AUC ↑ | F1-macro ↑ |
|---|---|---|
| SimCLR (Chen et al., 2020) - 5% | 0.735 | 0.547 |
| BYOL (Grill et al., 2020) - 5% | 0.752 | 0.564 |
| CLOCS (Kiyasseh et al., 2021) - 5% | 0.765 | 0.581 |
| LEAVES (Yu et al., 2022a) - 5% | 0.760 | 0.577 |
| METS (Li et al., 2023a) - 0% | - | 0.593 |
| MERL (Liu et al., 2024) - 0% | 0.757 | - |
| (*Ours*) ESI - 0% | **0.812** | **0.654** |

identification. Similar to the arrhythmia diagnosis task in Section 4.3, we focus solely on the test splits of the PTB-XL and ICBEB datasets. For PTB-XL, we select 5-second signal sequences from each of the 1907 subjects for both training and testing, which results in a 1907-class classification task. Similarly, we select 4-second samples and classes from the ICBEB dataset, which forms a 689-class classification task.

In this task, we compare the performance of the proposed ESI method with various approaches under linear probing and fine-tuning settings. For the specialized supervised methods, we compared our method with established supervised learners, including LSTM, XResNet101, ResNet50, and ensemble methods (Strodthoff et al., 2020). Besides the supervised learning methods, we also evaluated the ESI method against signal-focused SSL approaches including SimCLR (Chen et al., 2020), BYOL (Grill et al., 2020), CLOCS (Kiyasseh et al., 2021), and LEAVES (Yu et al., 2022a) under the linear probing and fine-tuning settings. Due to the patient de-identification during pretraining, we cannot perform zero-shot learning for this identification task, as the model has not been exposed to identifiable patient information.

Table 4 summarizes the findings. The results demonstrate that the ESI method significantly outperforms the baselines in both linear probing and fine-tuning settings. Notably, the fine-tuned ESI method achieves a substantial accuracy improvement (12.0% and 12.2% on PTB-XL and ICBEB, respectively) compared to the supervised ConvNeXt baselines. Additionally, in linear probing, ESI surpasses the signal-focused SSL methods by a significant margin. These findings highlight the effectiveness of multimodal pretraining with ECG and text data in learning transferable representations for ECG identification.

## 5 Discussion

In this section, we first discuss the ablations on the designed components, including the effectiveness of CQA, the selections of the signal encoder, as well as the contributions from captioning loss and contrastive loss to the learned representations. The experiments for ablations are mostly conducted on a tiny model variant as ESI-tiny with the ConvNeXt-tiny signal backbone. We also explore the impact of the size of pretraining data as well as the potential misalignment to the proposed method.

### 5.1 Ablation Study: Component Analysis and Impact

We conduct ablation experiments to assess the contributions of individual components within the pretraining framework. We evaluate the pretrained models' performance on both arrhythmia diagnosis and ECG-based user identification tasks using the PTB-XL dataset with the ESI-tiny variant featuring a ConvNeXt-tiny signal encoder. The model is evaluated under a linear probing setting with frozen encoders, and the AUC score serves as the primary metric.

The results of this ablation study are summarized in Table 5(a). Removing the Contrastive Question Answering (CQA) module results in a decrease in AUC scores for both tasks. This indicates that CQA contributes to aligning ECG signals with their enriched text annotations. By aligning these modalities, CQA helps the

Table 4: Evaluation results of 1-shot ECG-based user identification task under settings including supervised learning, linear probing, and fine-tuning. The metric used is the area under the ROC curve (AUC) and accuracy score (ACC). The best results are highlighted in **bold**.

| Method | PTB-XL | | ICBEB | |
|---|---|---|---|---|
| | $AUC \uparrow$ | $ACC \uparrow$ | $AUC \uparrow$ | $ACC \uparrow$ |
| *(Supervised)* | | | | |
| LSTM (Strodthoff et al., 2020) | $0.907_{\pm 0.014}$ | $0.444_{\pm 0.012}$ | $0.918_{\pm 0.011}$ | $0.610_{\pm 0.011}$ |
| XResNet101 (Strodthoff et al., 2020) | $0.915_{\pm 0.011}$ | $0.473_{\pm 0.015}$ | $0.921_{\pm 0.017}$ | $0.623_{\pm 0.020}$ |
| ResNet50 (Strodthoff et al., 2020) | $0.926_{\pm 0.016}$ | $0.497_{\pm 0.021}$ | $0.933_{\pm 0.013}$ | $0.641_{\pm 0.015}$ |
| Ensemble (Strodthoff et al., 2020) | $0.930_{\pm 0.011}$ | $0.500_{\pm 0.015}$ | $0.937_{\pm 0.013}$ | $0.653_{\pm 0.016}$ |
| ConvNeXt-Tiny (Woo et al., 2023) | $0.918_{\pm 0.014}$ | $0.480_{\pm 0.018}$ | $0.936_{\pm 0.014}$ | $0.647_{\pm 0.016}$ |
| ConvNeXt-Base (Woo et al., 2023) | $0.922_{\pm 0.013}$ | $0.494_{\pm 0.021}$ | $0.932_{\pm 0.013}$ | $0.649_{\pm 0.015}$ |
| *(Linear Probing)* | | | | |
| SimCLR (Chen et al., 2020) | $0.806_{\pm 0.022}$ | $0.185_{\pm 0.033}$ | $0.838_{\pm 0.019}$ | $0.252_{\pm 0.021}$ |
| BYOL (Grill et al., 2020) | $0.837_{\pm 0.019}$ | $0.240_{\pm 0.027}$ | $0.855_{\pm 0.019}$ | $0.283_{\pm 0.035}$ |
| CLOCS (Kiyasseh et al., 2021) | $0.822_{\pm 0.022}$ | $0.207_{\pm 0.036}$ | $0.841_{\pm 0.020}$ | $0.235_{\pm 0.037}$ |
| LEAVES (Yu et al., 2022a) | $0.840_{\pm 0.017}$ | $0.248_{\pm 0.029}$ | $0.853_{\pm 0.022}$ | $0.275_{\pm 0.029}$ |
| *(Ours)* ESI-tiny | $0.923_{\pm 0.014}$ | $0.510_{\pm 0.017}$ | $0.937_{\pm 0.012}$ | $0.654_{\pm 0.016}$ |
| *(Ours)* ESI | $0.927_{\pm 0.011}$ | $0.517_{\pm 0.014}$ | $0.944_{\pm 0.010}$ | $0.665_{\pm 0.012}$ |
| *(Fine-tune)* | | | | |
| SimCLR (Chen et al., 2020) | $0.936_{\pm 0.011}$ | $0.522_{\pm 0.014}$ | $0.945_{\pm 0.012}$ | $0.673_{\pm 0.015}$ |
| BYOL (Grill et al., 2020) | $0.942_{\pm 0.011}$ | $0.547_{\pm 0.014}$ | $0.951_{\pm 0.013}$ | $0.686_{\pm 0.012}$ |
| CLOCS (Kiyasseh et al., 2021) | $0.929_{\pm 0.009}$ | $0.516_{\pm 0.010}$ | $0.940_{\pm 0.011}$ | $0.666_{\pm 0.010}$ |
| LEAVES (Yu et al., 2022a) | $0.944_{\pm 0.011}$ | $0.550_{\pm 0.013}$ | $0.953_{\pm 0.009}$ | $0.688_{\pm 0.015}$ |
| *(Ours)* ESI-tiny | $0.966_{\pm 0.010}$ | $0.591_{\pm 0.014}$ | $0.980_{\pm 0.007}$ | $0.747_{\pm 0.012}$ |
| *(Ours)* ESI | $\mathbf{0.970}_{\pm 0.009}$ | $\mathbf{0.608}_{\pm 0.013}$ | $\mathbf{0.985}_{\pm 0.008}$ | $\mathbf{0.762}_{\pm 0.010}$ |

Table 5: Ablation experiments. The evaluation performances on the PTB-XL dataset for both arrhythmia diagnosis and identification tasks. The experiments are under the linear probing setting with frozen encoders, and the evaluation metrics used are the AUC scores. In the following tables, the performances from the most contributing components are highlighted in **bold**.

(a) Components of the proposed framework

| w/o | Arrhythmia | Identification |
|---|---|---|
| - | 0.928 | 0.923 |
| CQA | 0.901 | 0.902 |
| $\mathcal{L}_{\text{Cap}}$ | 0.913 | 0.905 |
| $\mathcal{L}_{\text{Con}}$ | **0.850** | **0.877** |

(b) Selection of the signal encoder

| Backbone | Params | Arrhythmia | Identification |
|---|---|---|---|
| ViT-1D | 84.92 M | 0.897 | 0.910 |
| XResNet101-1D | 1.80 M | 0.894 | 0.901 |
| ConvNeXt-tiny | 26.81 M | 0.928 | 0.923 |
| ConvNeXt-base | 85.56 M | **0.932** | **0.926** |

Table 6: Ablation experiment on pretraining ESI-Tiny on the MIMIC-IV-ECG data only. The table compares the performance of the model pre-trained exclusively on MIMIC-IV-ECG with the model pre-trained on the full dataset across different tasks and datasets.

| Pretraining Set | Arrhythmia (Linear Probing) | | Arrhythmia (Zero-shot) | | Identification (Linear Probing) | |
|---|---|---|---|---|---|---|
| | PTB-XL | ICBEB | PTB-XL | ICBEB | PTB-XL | ICBEB |
| MIMIC | 0.912 | 0.967 | 0.792 | 0.832 | 0.917 | 0.934 |
| All | 0.928 | 0.976 | 0.807 | 0.842 | 0.923 | 0.937 |
| *diff* | ↓ 0.016 | ↓0.009 | ↓0.015 | ↓ 0.010 | ↓ 0.006 | ↓ 0.003 |

model learn more robust representations that capture the inherent relationships between ECG patterns and associated diagnoses. Ablating the captioning loss $\mathcal{L}_{\text{Cap}}$ also leads to a performance decline on both arrhythmia diagnosis and ECG identification, which suggests that the model benefits from explicitly generating captions during pretraining. The most substantial impact on performance is observed when removing the contrastive loss $\mathcal{L}_{\text{Con}}$. By contrasting ECG with different textual representations, the model is encouraged to identify subtle variations that are relevant to downstream tasks.

## 5.2 Ablation Study: Impact of Pre-Training with MIMIC-IV-ECG Only

As the MIMIC-IV-ECG data is the single dataset that contains the most ECG samples used in pretraining our ESI models, to investigate the impact of using only the MIMIC-IV-ECG dataset for pre-training, we conducted an experiment where the model was pre-trained exclusively on MIMIC-IV-ECG. We then evaluated the model's performance on the PTB-XL and ICBEB datasets, focusing on three tasks: arrhythmia detection (both linear probing and zero-shot) and subject identification (linear probing).

Table 6 presents the performance results across these tasks. The results indicate that pre-training on the full dataset yields better performance across all tasks and datasets compared to pre-training exclusively on MIMIC-IV-ECG. Specifically, there is a noticeable improvement in both arrhythmia detection and subject identification tasks when additional datasets are included in the pre-training process. For the arrhythmia detection task, the difference in performance is more substantial in the zero-shot setting, where the model pre-trained on all datasets achieves an improvement of 0.015 in PTB-XL and 0.014 in ICBEB. In the identification task, the model pre-trained on all datasets only shows a slight improvement.

These results suggest that while MIMIC-IV-ECG provides a strong foundation for pre-training, and the inclusion of additional datasets can further enhance the model's ability to generalize across different tasks and datasets. Also, for the zero-shot learning tasks, learning from the samples and annotating information from the same dataset during pretraining, e.g., PTB-XL, could help the downstream tasks accordingly.

## 5.3 Ablation Study: Selection of Backbone for Signal Encoder

We investigate the impact of various backbone architectures for the signal encoder on the model's performance. Prior research on vision-text pretraining informed our selection of candidate backbones, including

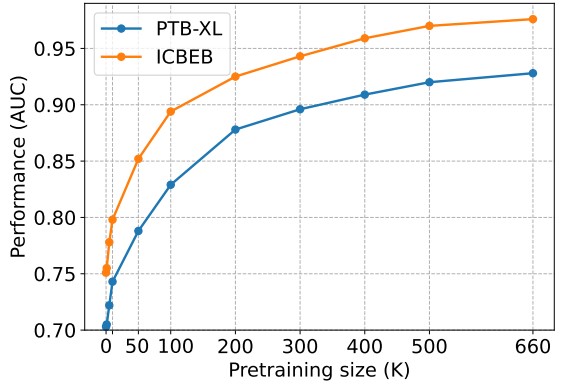

Figure 5: Performances of linear probing inference in arrhythmia diagnosis (AUC) on PTB-XL and ICBEB data using the pretrained encoders with varying training samples.

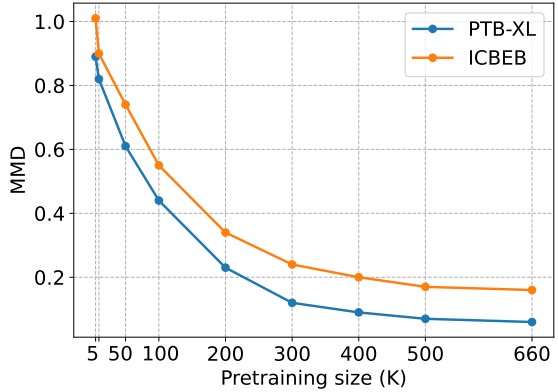

Figure 6: Distribution difference measured by maximum mean discrepancy (MMD) between the pretraining set and test set using encoders with varying training samples.

ViT (Dosovitskiy et al., 2020) and ConvNeXt (Woo et al., 2023) architectures. Additionally, considering the strong performance of XResNet1D101 in supervised arrhythmia diagnosis (Strodthoff et al., 2020) as shown in Table 2, we examine its potential as a foundation encoder after multimodal pretraining.

Table 5(b) summarizes the results. ConvNeXt-based models (ConvNeXt-tiny and ConvNeXt-base) achieved the highest AUC scores on both arrhythmia diagnosis and ECG identification tasks. Notably, ConvNeXt-base, the largest model with the most parameters (85.56M), provides the best overall performance (AUC of 0.932 for arrhythmia diagnosis and 0.926 for ECG identification). ConvNeXt-tiny, a more parameter-efficient option (26.81M), shows a slightly lower performance compared to the base version, but also achieves competitive AUC scores with a significantly lower parameter footprint. With the same-level size as ConvNeXt models in both tasks, the ViT model shows substantially lower performances, which might indicate the convolutional kernel could be more suitable for processing ECG signals. Moreover, the XResNet1D101 backbone displays lower AUC scores compared to ConvNeXt architectures. This can be due to the significantly smaller parameter size of the XResNet model.

### 5.4 Ablation Study: Impact of Pretraining Data Sizes and Distribution Shifts

In this ablation study, we also investigate the impact of pretraining dataset size on the performance of our ESI-tiny model in two arrhythmia classification datasets, including PTB-XL and ICBEB. We artificially change the size of the unlabeled pretraining dataset from 0 to all available samples by randomly sampling and selecting from all the pretraining samples. After ESI-tiny encoders are trained, the models are then evaluated on both datasets using a linear probing approach with frozen encoder weights, and the AUC score is employed as the performance metric.

The results presented in Figure 5 demonstrate a consistent performance improvement as the pretraining dataset size increases for both PTB-XL and ICBEB datasets. The model's performance improves most substantially in the early stages of increasing the pretraining dataset size. This indicates that a modest amount of pretraining data can help substantial gains in learning informative representations.

Additionally, to understand the effect of pretraining on distribution shift, we measure the Maximum Mean Discrepancy (MMD) between the pretraining dataset and each test set (PTB-XL and ICBEB) as the pretraining size increases. As shown in Figure 6, the distribution shift between the pretraining and test sets decreases as the pretraining dataset size increases. This suggests that larger pretraining datasets help the model learn representations that are more generalizable to the target datasets. The decrease in MMD correlates with the improvements in AUC scores observed in Figure 5, which potentially indicates that reducing distribution shift through larger pretraining sets contributes to better performance on the downstream tasks.

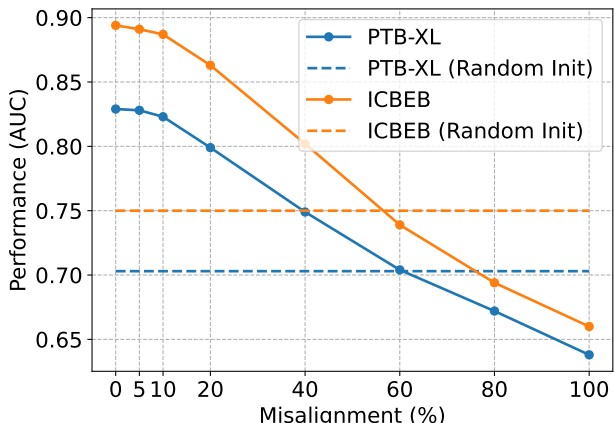

Figure 7: Performances of linear probing inference in arrhythmia diagnosis (AUC) on PTB-XL and ICBEB data using pretrained encoders with varying ECG-text misalignment ratios. Models trained with randomly initialized weights are included as baselines.

### 5.5 Ablation Study: Impact of ECG-text Misalignment

To investigate the impact of potential misalignments between ECG signals and their corresponding text descriptions on the proposed ESI method, we conduct an experiment on ESI-tiny with a sub-training set of 100K ECG-text pairs. We introduce various degrees of misalignment by randomly shuffling a percentage of ECG-text pairs in our pretraining dataset. The model was then evaluated on two arrhythmia classification datasets with a linear probing approach with frozen encoder weights.

Figure 7 shows the results of the experiment. As the percentage of misaligned pairs increases, the performance of ESI-tiny decreases substantially on both datasets. This indicates that the alignment between the pretraining sample pairs plays an essential role in learning robust representations. Notably, increasing misalignment can lead to performance that is worse than using un-pretrained models with random initialization, which also highlights the critical importance of accurate ECG-text pairing in pretraining.

## 6 Conclusion

This study introduces a novel multimodal contrastive pretraining framework to enhance the quality and robustness of representations learned from ECG signals. To address the lack of descriptive text associated with ECGs, we propose a retrieval-augmented generation (RAG) pipeline called the Cardio Query Assistant (CQA). This pipeline generates detailed textual descriptions for ECG data with demographic information, potential conditions, and waveform patterns. Inspired by the success of multimodal pretraining strategies in vision-language tasks, we develop the ECG Semantic Integrator (ESI). This framework integrates both contrastive and captioning capabilities to foster a deeper semantic understanding of ECG signals. Our evaluation validates the effectiveness of the proposed approach across several downstream tasks. The ESI method demonstrates improvement in arrhythmia diagnosis and ECG-based user identification tasks by outperforming strong baselines that cover supervised learning and SSL approaches. These results with the ablation studies highlight the benefits of multimodal learning for ECG analysis and the value of integrating captioning loss with contrastive pretraining. Beyond ECG, we believe the proposed CQA and ESI frameworks hold the potential for applications to other types of biomedical time series data, where contextual information can be leveraged to enhance the representation learning and downstream analysis.

On the other hand, this study is limited by the use of 10-second ECG signals only in pretraining. While our approach demonstrates effectiveness on this data, real-world ECG recordings can vary significantly in length and may contain more diverse features. In future work, we plan to investigate the impact of using more diverse ECG signals on the performance of the proposed framework.

**Acknowledgments**

This work was supported by National Science Foundation (# 2047296) and National Institute of Health (# R01DA059925)

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

# A   Ablation Study on CQA

The introduction of Retrieval-Augmented Generation (RAG) in CQA in this study plays a crucial role in enriching the pretraining dataset by generating detailed ECG waveform descriptions. In this ablation study, we investigate the impact of different RAG settings on performance.

To evaluate the performance of our RAG-generated descriptions, we utilize the human-annotated ECG waveform descriptions from (Wen & Kang, 2022) as references. This dataset includes annotations for 28 types of arrhythmia. We generate corresponding descriptions using our RAG method and compared them against the reference descriptions using BERTScore (Zhang et al., 2020) with the *RoBERTa-large* model (Liu et al., 2019). BERTScore is a metric designed to evaluate the similarity between generated text and reference text by leveraging pre-trained contextual embeddings from transformer models. BERTScore computes the similarity of embeddings to provide a more nuanced assessment of semantic alignment, and there are three main metrics served in BERTScore:

- **Precision** measures how well the tokens in the generated text match with the tokens in the reference text, which indicates how accurately the generated content reflects the reference.

- **Recall** assesses how well the tokens in the reference text are captured by the generated text, which reflects the completeness of the generated content.

- **F1 Score** is the harmonic mean of Precision and Recall, which provides a balanced measure of both accuracy and completeness in the semantic alignment between the generated and reference texts.

## A.1   Impact of Top-k in Vector Search

In this section, we examine the effect of varying the top-k parameter in the vector search interface on the model's performance. The top-k parameter controls how many of the top-ranked candidates are considered in the generation process. We evaluated the model's performance by measuring Precision, Recall, and F1 score generated by BERTScore for different values of top-k. The mean and standard deviation of these metrics across multiple runs are summarized in Table 7.

Table 7: BERTScores across different Top-k values

| Top K | $\mathbf{B}_{Precision}$ ↑ | $\mathbf{B}_{Recall}$ ↑ | $\mathbf{B}_{F1Score}$ ↑ |
|---|---|---|---|
| 1 | 0.8190 (0.0128) | 0.8326 (0.0237) | 0.8256 (0.0147) |
| 2 | 0.8197 (0.0159) | 0.8402 (0.0156) | 0.8296 (0.0090) |
| 3 | 0.8169 (0.0160) | 0.8433 (0.0155) | 0.8297 (0.0106) |
| 4 | 0.8159 (0.0157) | 0.8447 (0.0160) | 0.8299 (0.0111) |
| 5 | 0.8153 (0.0172) | 0.8447 (0.0167) | 0.8295 (0.0110) |

The results indicate that as the top-k value increases, the Recall generally improves slightly, which suggests that considering more candidates helps in retrieving more relevant information. However, the mean Precision tends to decrease marginally, which reflects a trade-off where including more candidates might introduce less relevant information. The F1 score, which balances both Precision and Recall, remains relatively stable across different top-k values. This indicates that the model maintains a consistent performance overall.

Our findings suggest that varying the top-k parameter has a limited but consistent impact on RAG performance. The choice of top-k can be adjusted based on the specific needs of the task, with higher values favoring recall at a slight cost to precision, while lower values provide a more precise but slightly less comprehensive output. In this study, we set the top-k value as 2.

## A.2   Impact of the Number of Textbooks in RAG

We investigate the impact of the number of textbooks used as the knowledge base in the CQA pipeline. The purpose of this ablation study is to assess whether the inclusion of more or fewer textbooks affects the

Table 8: BERTScores across different numbers of textbooks used in RAG

| Textbooks | $\mathbf{B}_{Precision}\uparrow$ | $\mathbf{B}_{Recall}\uparrow$ | $\mathbf{B}_{F1Score}\uparrow$ |
|---|---|---|---|
| No Textbook | 0.7617 (0.0187) | 0.8154 (0.0142) | 0.7928 (0.0126) |
| Garcia (2015) | 0.8169 (0.0142) | 0.8290 (0.0175) | 0.8217 (0.0111) |
| Huff (2006) | 0.8182 (0.0145) | 0.8395 (0.0170) | 0.8285 (0.0100) |
| Both | 0.8197 (0.0159) | 0.8402 (0.0156) | 0.8296 (0.0090) |

quality of the generated ECG waveform descriptions. The textbooks used in this study are *"ECG Workout: Exercises In Arrhythmia Interpretation"* by Huff (2006) and *"12-Lead ECG: The Art of Interpretation"* by Garcia (2015). We conducted experiments using three different configurations: using no textbooks, using each textbook individually, and using both textbooks combined. The results are summarized in Table 8.

The results show a clear trend that incorporating textbooks into the RAG pipeline improves the performance across all BERTScore metrics (Precision, Recall, and F1 Score). Specifically, using both textbooks in combination yields the highest scores, with a Precision of 0.8197, a Recall of 0.8402, and an F1 score of 0.8296. This suggests that the inclusion of multiple sources of knowledge enriches the generated descriptions. When using only one textbook, Huff (2006) slightly outperforms Garcia (2015) in terms of all three metrics, although the difference is incremental. This indicates that while each textbook independently contributes to the generation quality, combining them provides a more robust knowledge base for the RAG process. In contrast, the absence of textbooks results in a significant drop in performance, with the lowest scores across all metrics. This highlights the importance of incorporating domain-specific knowledge to improve the relevance and accuracy of LLMs in generating content. Thus, in this study, using multiple textbooks in the RAG pipeline not only increases the quality of the generated ECG descriptions but also ensures consistency in the output.

### A.3 Impact of Different LLM Prompting Strategies

In this subsection, we investigate the effect of varying the prompt structure used in the LLM within the RAG pipeline on the quality of the generated ECG waveform descriptions. To assess consistency of the generation across different prompts, we conducted an ablation study by rephrasing our standard prompt, "How is *{ECG condition}* reflected in 12-lead ECG?", into five different variations such as "Describe how *{ECG condition}* appears in a 12-lead ECG." and "What are the ECG findings associated with *{ECG condition}* in a 12-lead setup?" The alternative prompts used are designed to explore whether slight changes in wording would affect the performance of the RAG pipeline. Despite these variations, the underlying query remains focused on generating descriptions of how specific ECG conditions are in a 12-lead ECG.

The results demonstrate a remarkable consistency across all rephrased prompting strategies, with an average mean F1 score of 0.8295 and a very low standard deviation of 0.0003 across the variations of prompts. This indicates that the CQA's ability to generate relevant descriptions is robust to variations in the phrasing of the prompt. This robustness allows for some flexibility in how prompts are formulated without hurting the quality of the generated content.

## B Datasheet for ECG Datasets Used for Training ECG Model

In this section, we compile a dataset for the used training data following Gebru et al. (2021).

### B.1 Motivation

**Purpose**: The datasets were collected and processed to facilitate the development and evaluation of automated ECG processing algorithms, particularly for training a foundation model that extracts generalized embedding from ECG signals. These datasets provide a large number of ECG recordings paired with human or machine annotations and metadata, which enables robust training of machine learning models in the medical domain.

**Creators**: The datasets were created by institutions including PhysioNet, Chapman University, Shaoxing People's Hospital, and Physikalisch-Technische Bundesanstalt (PTB).

**Funding**: Development and release of these datasets were supported by various grants, including the NIH, the Kay Family Foundation, and the Berlin Big Data Center.

### B.2 Composition

**Instances**:

- **MIMIC-IV-ECG**: Contains diagnostic 12-lead ECG recordings, with each instance representing an ECG signal paired with machine-generated annotations and metadata.

- **PTB-XL**: Comprises 17,415 clinical 12-lead ECG records, annotated by cardiologists, to cover a wide range of diagnostic categories.

- **Chapman-Shaoxing**: Includes 45,152 12-lead ECGs from patients, annotated by professional experts, with conditions such as atrial fibrillation, myocardial infarction, and other arrhythmias.

**Volume**:

- **MIMIC-IV-ECG**: The dataset includes thousands of ECG records.

- **PTB-XL**: Contains 17,415 ECG records.

- **Chapman-Shaoxing**: Contains 45,152 ECG records.

**Data Features**: Each dataset includes ECG waveform data, demographic information (age, sex, etc.), and clinical annotations related to cardiac conditions. PTB-XL and Chapman-Shaoxing datasets also include metadata on the diagnostic process, such as SCP-ECG statements.

### B.3 Collection Process

**Data Acquisition**:

- **MIMIC-IV-ECG**: Data was collected as part of routine clinical care and stored in the MIMIC-IV database.

- **PTB-XL**: ECG signals were collected using devices from Schiller AG over a period of seven years, with annotations provided by cardiologists.

- **Chapman-Shaoxing**: Data was collected from multiple hospitals, with ECG signals recorded and labeled by licensed physicians.

**Ethical Considerations**: All datasets are publicly available and de-identified to protect patient privacy. The collection of the original data was approved by relevant institutional ethics committees. No new data was collected in this study.

### B.4 Preprocessing/Cleaning/Labeling

**MIMIC-IV-ECG**: Machine-generated annotations were used for labeling. The data was cleaned to remove noise and artifacts.

**PTB-XL**: ECG data was preprocessed to ensure consistency in the waveform format, and SCP-ECG codes were applied for standardized labeling.

**Chapman-Shaoxing**: ECG data was denoised, and diagnostic information was verified by multiple physicians before being stored in a structured format.

### B.5 Uses

**Current Uses**: These datasets have been used for developing and evaluating machine learning models for ECG analysis, particularly in pretraining ECG models and detecting arrhythmias.

**Potential Uses**: The datasets can be used for broader research in cardiac health, including developing diagnostic tools and studying the epidemiology of heart conditions.

**Restrictions**: While the datasets are de-identified, users should still adhere to ethical guidelines when using the data for research.

### B.6 Distribution

**Access**: The datasets are publicly available through the PhysioNet platform under the Creative Commons Attribution 4.0 International License.

**Regulatory Issues**: There are no specific regulatory restrictions, but users must follow the licensing terms.

### B.7 Maintenance

**Support**: Users can contact the dataset maintainers through the PhysioNet platform for support or questions.

**Updates**: Any updates or new versions of the datasets will be announced on the PhysioNet platform.

**Community Contributions**: Researchers are encouraged to contribute their findings and improvements back to the community through publications or direct contributions to the dataset repository.

