# OpenReview forum: "ECG Semantic Integrator (ESI): A Foundation ECG Model Pretrained with LLM-Enhanced Cardiological Text"
_TMLR — Accepted by TMLR_

### Review · Reviewer_QuP6 · 2024-06-25

**Summary Of Contributions:**

This paper studies the multi-modal pretraining for ECG representation models. It considers waveforms and their text descriptions, and the original text descriptions of the waveforms are enhanced with an LLM-based RAG pipeline. The multi-modal training combines a contrastive loss and a captioning loss. Experiment results show that the proposed method improves the accuracy of existing approaches for downstream tasks.

The main contribution of the paper is using an LLM-based RAG pipeline to enhance the text descriptions.

**Audience:**

Yes

**Broader Impact Concerns:**

The paper uses machine learning for medical purposes and may benefit public health. I can see no negative impacts for this paper.

**Claims And Evidence:**

Yes

**Requested Changes:**

I think the authors have shown that using LLM-based RAG pipeline to enhance the text descriptions works. For a revision, the authors should make the following changes to address the weakness.

R1: Report standard deviation for the accuracy results.

R2: Thoroughly explore the design space of LLM-based RAG and do a better job in justifying the designs.

**Strengths And Weaknesses:**

Strength

S1: Using LLM-based RAG pipeline to enhance the text descriptions makes senses and aligns with researches that use LLM to reduce labeling efforts.

S2: The model design follows SOTA methods and yields good experiment results.

S3: The experiment results are comprehensive.

S4: The presentation is clear and easy to follow.

============

Weakness

W1: Please report standard deviation for accuracy by running several random initializations (e.g., 10). Currently, the accuracy improvement is marginal in some cases (which is OK), and it is unclear whether these improvements are caused by randomness.

W2: The core contribution of the paper, i.e., LLM-based RAG pipeline, is not explored in depth. For instance, vector search adopts a top-k interface, and how does the value of k affect accuracy? Currently, two textbooks are used as the knowledge base for vector query, will the accuracy be affected if more or fewer textbooks are used? Is it possible that the textbooks return in-consistent contents for the same vector query? How does the authors prompt the LLM? Do different ways of prompting the LLM affect accuracy?

---

> ### Author Response · Authors · 2024-08-21
> **Thank you reviewer QuP6 for your insightful comment**
>
> Thank you for your detailed and constructive feedback on our paper. We have carefully considered your suggestions and made the following updates to address the concerns raised:
>
> W1: We have added the standard deviations for the accuracy results in the main results section of the paper. By running several random initializations, we ensured that the reported accuracy improvements are robust and not merely due to randomness. These additions provide a clearer understanding of the consistency and reliability of our model's performance across different runs.
>
> W2: We have conducted a thorough ablation study to explore the design space of the LLM-based RAG pipeline as suggested. The new ablation and discussion have been added to Appendix as A.1. Specifically, we investigated the following:
>
> - The impact of the top-k parameter in vector search on model performance. The results of this study, including detailed discussions, have been added to the appendix for further reference.
>
> - The effect of using different numbers of textbooks as the knowledge base for the vector query. We found that incorporating more textbooks generally improves performance, as detailed in the appendix.
>
> - The impact of different prompting strategies on the quality of the generated text. We experimented with five different rephrased prompts and found that the performance remains consistent across all variations, with minimal differences in accuracy. These findings are also included in the appendix.
>
> We hope that these revisions address your concerns and provide a more comprehensive understanding of the contributions and design choices in our work. Again, we appreciate your valuable feedback in helping us improve our submission.

---

### Review · Reviewer_yd5W · 2024-06-28

**Summary Of Contributions:**

By arguing that multimodal representation learning is less explored for ECGs and self-supervising solely based on signals is insufficient for exploring the underlying semantic information, this paper proposed a retrieval-augnmented generation (RAG) pipeline to generate a large ECG-text pairs by leveraging knowledge extracted from two books and the LLM. The generated dataset of ECG-text pairs (650k signals) was then used for pretraining based on a particular contrastive learning method. Various ablation studies were performed to evaluate its robustness against a number of training related factors, e.g., the changes in size of pre-training data, degree of ECG-text mis-alignment, etc.

The key contribution of this work lies in showing the possibilty of multi-modal representation learning by incorporating prior medical knowledge and evaluating how well the pre-trained can perform for the downstream tasks under various settings (zero-shot, few-shot, linear probing and fine-tunng).

**Audience:**

Yes

**Broader Impact Concerns:**

As publicly available anonymized datasets are used for the study, I do not have concerns on the ethical implications of this work.

**Claims And Evidence:**

Yes

**Requested Changes:**

**Specific comments:**
- Some typos spotted in Figure 1, e.g., “kownledge”, “accosicates”.
- Section 3.1.2 provides the general ideas about the use of pre-constructed domain-knowledge database, given information such as standard clinical labels, standard communications protocol for computer-assisted ECG (SCP) statements, diagnostic interpretations, and machine-generated reports associated with ECG data. Some descriptions of the different types of information and how they are stored in the database should be included. Also, more details about how to generate the waveform description should be included.
- One of the books is “ECG Workout: Exercises In Arrhythmia Interpretation by Huff”. Will this source of the information be biased for the downstream task?
- PTB-XL and Chapman-Shaoxing are included for pre-training and evaluation, and MMIC-IV-ECG is used only for pre-training. What happens if only MIMIC-IV-ECG is used for pre-training? Under this setting, it will be interesting to see how the zero-shot and linear-probing performance on the two other datasets will be affected. Alternatively, can some of the MIMIC-IV-ECG data be hold out for evaluation as well?
- The sizes of the three datasets are very different, and MIMIC-IV-ECG obviously dominates the pre-training. Also, is there any class imbalanced situation for the three datasets?
- According to Table 5a, removing CQA module will lead to performance drop in AUC but not really a big drop. Please contrast the performance gained by incorporating CQA as compared to putting further effort in collecting more data. Under what situations will the performance gain be more obvious?

**Strengths And Weaknesses:**

**Strengths:**
- Demonstrated an effective RAG pipeline for incorporating prior knowledge in text to address the challenge of lacking annotated data (particularly in medical field)
- Developed a foundation model based on a dataset with 650k paired data items via constrastive learning
- Carefully evaluated the foundation model (and also a tiny version in view of the possibility of limited computing resources).

**Weaknesses:**
- The tricks being adopted are inspired from some other works as explained in the paper. The novely regarding machine learning algorithms is limited.
- While the paper organization and presentaiton is clear in general, some background about some details of ECG (e.g., SCP is not much explained in the manuscript) could be added to further improve the readability for those without bg.

---

> ### Author Response · Authors · 2024-08-21
> **Thank you reviewer yd5W for your insightful comment**
>
> Thank you for your thorough review and valuable feedback on our paper. We have carefully addressed your comments and made the following revisions:
>
> 1. **Typos in Figure 1:** We have corrected the typos in Figure 1 and thoroughly reviewed the rest of the text and figures to ensure there are no other errors.
>
> 2. **Enhancements to Section 3.1.2:** We have added more detailed descriptions in Section 3.1.2 regarding how waveform information is queried from the pre-constructed domain knowledge database. Brief descriptions of ECG-related knowledge, such as the SCP statements, have been included. Additionally, we have provided more details on how the waveform descriptions are generated.
>
> 3. **Potential Bias from “ECG Workout: Exercises In Arrhythmia Interpretation by Huff”:** We appreciate your concern about potential bias introduced by using "ECG Workout: Exercises In Arrhythmia Interpretation by Huff" as one of the primary sources in our knowledge base. This book, which covers the use of 12-lead ECGs in diagnosing heart-related conditions, is indeed a valuable resource. To ensure its impact is beneficial, we conducted an ablation study (now included in Appendix 1) to evaluate the influence of using different textbooks within the RAG framework, including the use of "ECG Workout" alone, in combination with other sources, and without it. Our findings show that while "ECG Workout" enhances performance in waveform generation compared to using LLMs directly, and combining it with other sources does not negatively impact performance.
>
> 4. **Impact of Using Only MIMIC-IV-ECG for Pre-Training:** We conducted an ablation study (as shown in Section 5.2) where the model was pre-trained using only the MIMIC-IV-ECG dataset and then evaluated on the PTB-XL and ICBEB datasets. The results showed a modest drop in performance compared to using the full dataset for pre-training, which indicates that while MIMIC-IV-ECG alone provides a solid foundation, additional datasets improve generalization. Furthermore, removing PTB-XL from the pre-training set had a more substantial impact on zero-shot performance on the PTB-XL arrhythmia task.
>
> 5. **Hold out MIMIC-IV-ECG for Evaluation:** Thank you for your suggestion. We chose not to use MIMIC-IV-ECG as part of the evaluation set because it contains only machine-generated annotations. Without human-validated annotations, it might not serve as an evaluation set.
>
> 6. **Dataset Sizes and Class Imbalance:** We acknowledge that the MIMIC-IV-ECG dataset is the largest among the three, which allows it to dominate the pre-training process. We selected this dataset as it is one of the largest publicly available sources. To manage this, we incorporated other datasets into the pre-training process. Regarding class imbalance, it is indeed present in the PTB-XL and Chapman-Shaoxing datasets, with normal ECG conditions being the most prevalent. We addressed this by following evaluation procedures from prior ECG arrhythmia-focused studies and maintaining consistency in our evaluation settings.
>
> 7. **Performance Gain from Incorporating CQA:** Thank you for your observation regarding the impact of removing the CQA module. The CQA module provides detailed waveform descriptions, and its removal leads to only a moderate drop in AUC because the model still benefits from other annotations such as demographic information and ECG condition labels. The CQA module serves as an additional form of supervision to enhance the model's understanding of the data. Based on our ablation studies, including pre-training with only MIMIC-IV-ECG and the study in Section 5.4, we found that reducing the amount of pre-training data (e.g., using only MIMIC-IV-ECG) does not result in a significant performance drop. Collecting more data can improve the model's ability to learn from a broader range of samples, while incorporating CQA adds value by introducing semantic information related to ECG, which can also be beneficial in enhancing model performance.

---

### Review · Reviewer_8LVf · 2024-07-31

**Summary Of Contributions:**

This paper introduces a contrastive pretraining approach for multimodal data (ECG signals and text) as well as a RAG-based approach to create diverse training datasets that serve as pre-training data for the contrastive approach. They show that when using their method to create a pretraining dataset that is then used with their method + objective function for training, this can learn generalisable representations for ECG, as evaluated on several downstream tasks (e.g., arrhythmia detection) in the medical domain. They also show that this improves over baselines that have more supervision—both supervised and self-supervised methods.

**Audience:**

Yes

**Broader Impact Concerns:**

None.

**Claims And Evidence:**

Yes

**Requested Changes:**

1. Add an overview of the current state-of-the-art approaches (post CLIP) in the related works section. Even if this paper uses CLIP as the main architecture (which is fine!) there have been several works since then that propose different pre-training strategies that could be cited. See examples above and from here. https://arxiv.org/pdf/2202.09061
2. Add a paragraph that outlines the differences in the contrastive pretraining approach used here vs. these newer pretraining approaches and why the current approach is more favourable. This would make it easier for authors to understand the decisions made and why this architecture might be favourable for this particular data/waveforms as opposed to other architectures.
3. Add a detailed datasheet for the dataset produced? See the comment in weaknesses above and example datasheets here. https://arxiv.org/abs/1803.09010
4. Add either a figure showing readers what the data looks like (e.g., how the waveforms are represented, what a corresponding text snippet is, what demographics are) or a table with details of attributes of data in the section outlining the datasets.
5. For all results, link in the text where each result is shown. There are several cases in the paper where it is hard to find the table showing results explained in that paragraph and would make it easier to go back to that table (especially since they are all so close to each other in the evaluation section).
6. Add confidence metrics and resample from the model at least 5 times to assess robustness of model generations.

**Strengths And Weaknesses:**

Strengths:


1. This paper is clearly written and explained and tackles an important problem in the medical domain to build multimodal pretrained representations for ECG diagnosis.
2. It is well situated in the literature both in the previous relevant multimodal models and the ECG-specific models that are currently state of the art. (note: I’ve pointed out some other works to be cited as well as newer methods to look into).
3. The experimental methodology is clearly explained and experimentally sound.
4. The evaluations compose of zero shot evaluations, linear probing and evaluations of fine-tuned components to downstream tasks which allow us to both look at model representations as well as the ability to generalise which is good to see.
5. This method shows improved performance over the previous best models (even when fully supervised) and that the fine-tuned ESI model outperforms all previous models when looking at AUC by a small margin.
6. They also show the robustness of the learned representations by evaluating that this generalises to downstream tasks better than previous models that learn multimodal representations.
7. Question: What is the demographic information in the dataset? Apologies if I missed this but it’s not present clearly in the section that describes the data and is an important component of it.
8. Question: What is the reasoning behind using a ConvNet encoder vs. another encoder (ResNet, or more complicated prefix-lm type encoders). The reasoning behind the encoder choices would be good to explain in the paper for future work to build off of/compare other encoders to if needed.
9. Question: Have the authors experimented with augmenting the generated data with perturbations, entailments etc., to expand the dataset?
10. Question: Have the authors evaluated the benefit of a RAG-based approach to other sampling methods for the dataset to create a pretraining dataset? It’s great to see good performance with RAG, but it would be good to verify that this is superior to other ways of retrieving from existing large resources to create a new dataset that you then evaluate by looking at model performance on this data.
11. Question: When using BioLinkBert for the text encoding it would be good to have a few sentences outlining how this is done and what is different from a standard BERT encoder (or a baseline that is not specific to biomedical data).

Weaknesses:

1. The introduced model/architecture is largely based on the CLIP model (Radford et. al., 2021) from a few years ago. While this is still a very competitive model and these representations do form the basis of the other stronger models that are currently SOTA, there have been other approaches (e.g., PrefixLM, VITs, fusion with cross attention https://arxiv.org/abs/2108.10904, https://arxiv.org/abs/2204.14198, https://arxiv.org/abs/2102.10407) in the last few years that have far surpassed this in multimodal pre-training.
2. The data generation and RAG-retrieval process is described  clearly but attributes of the data itself are not very clear. If the authors could add a table or figure that outlines the components of the data (waveform, demographics) with an example snippet from the dataset that would illustrate this very nicely. This coupled with a datasheet that gives us summary statistics of demographics, waveforms, text, any private information etc., would be good to ensure any safety concerns about the dataset as well.
3. A datasheet that outlines the data generated would be good to have. This would be particularly helpful for this dataset since (a) it could potentially concern private human information and (b) the exact form of the data might be unfamiliar to researchers given the various modalities, it would be nice to document all of this in one place for clarity. See this paper and example datasheets here. https://arxiv.org/abs/1803.09010
4. The metrics reported are largely just AUC for the three different categories (supervised, probing, fine-tuning) and we see a small difference of around 0.01 between the best performing models. To properly evaluate whether this effect is significant, it would be good to have accuracy measures with confidence scores, averages over at least 5 different samples of the model for every sample, and calibration scores of the model as well if possible.

---

> ### Author Response · Authors · 2024-08-21
> **Thank you reviewer 8LVf for your insightful comment (part 1)**
>
> Thank you for your thorough review and valuable feedback on our paper. We have carefully addressed your comments and made the following revisions:
>
> 1. **Overview of the current state-of-the-art multimodal pretraining approaches**:
> Thank you for your suggestion. We have updated and enriched Section 2.1 to include additional state-of-the-art image-text pretraining studies.
>
> 2. **Differences in the contrastive pretraining approach used here vs. newer pretraining approaches**:
> We appreciate your insightful feedback. We have added a paragraph in Section 3.2 that outlines our reasoning for selecting the pretraining approach used in this study. Specifically, we explain that our method draws inspiration from the proven methodologies of CLIP. We chose this approach because the core contrastive objective of CLIP is particularly well-suited to aligning ECG signals with their textual interpretations. Additionally, the simplicity of the CLIP architecture allows for seamless integration of captioning losses inspired by CoCa, which is proven to be beneficial in our study. We further validate the contribution of these objectives through ablation studies, as shown in Section 5.1.
>
> 3. **Detailed datasheet for the dataset**: Thank you for your suggestion. We have created and included a comprehensive datasheet in Appendix B. This datasheet provides an overview of the datasets used in our study, including their structure, attributes, and the key demographic information available. We believe this addition will enhance transparency and provide clarity for readers who may be less familiar with the specific datasets used in this research.
>
> 4. **Figure/Table showing dataset attributes**:
> To better illustrate the data we are working with, we have included a new figure (Figure 2) that presents an example of a raw ECG signal, its associated metadata, and the generated ECG waveform description.
>
> 5. **Linking results to tables**:
> Thank you for your comment. We have reviewed the manuscript and improved the references to tables throughout the text. This revision should ensure that each result is clearly linked to its corresponding table.
>
> 6. **Confidence metrics and resampling for robustness**:
> We have added standard deviation values to our reported results. These metrics reflect the robustness of our model performances, which were resampled at least five times to ensure that the results are consistent and reliable across different runs.
>
> 7. **Demographic information in the dataset**:
> Thank you for bringing this to our attention. We apologize for not clearly presenting the demographic information in the initial submission. The demographic information for all three datasets used in our study is limited to age and gender. This information is now clearly stated in the manuscript to ensure transparency and to help contextualize the dataset's composition.
>
> 8. **Reasoning behind using a ConvNet encoder vs. other encoders**:
> We appreciate your question regarding our choice of the ConvNet encoder. We have conducted a comparison of different backbone architectures, including ViT-1D, ResNet-1D, and ConvNeXt, as detailed in our ablation study (Section 5.3).
>     - **ViT-1D**: Although the ViT architecture has shown promise in many vision tasks, its linear embedding layers did not prove advantageous for processing ECG data. The performance was lower than that of ConvNeXt, which suggests that ViT’s structure may not be optimally suited for capturing the nuanced features in ECG signals.
>     - **ResNet-1D**: While the ResNet structure is known for its robustness, the version we tested (XResNet101-1D) was limited by its smaller parameter size. This constraint hindered its ability to scale effectively with the larger volume of ECG data.
>     - **ConvNeXt**: Both the tiny and base versions of ConvNeXt demonstrated superior performance, with ConvNeXt-base achieving the highest AUC scores after pretraining. This architecture's ability to capture both local and global contexts within ECG waveform data made it the most effective choice for our tasks.
>
>     Based on these results, we selected ConvNeXt for its performance and scalability.
>
> 9. **Augmenting the generated data**:
> Yes, we have indeed implemented data augmentation techniques during pretraining, following established practices in multimodal pretraining works. For example, we applied jittering noise (Gaussian noise with a standard deviation of 0.003) and scaling augmentation as default settings.

---

> > ### Author Response · Authors · 2024-08-21
> > **Thank you reviewer 8LVf for your insightful comment (part 2)**
> >
> > 10. **Evaluation of the RAG-based approach compared to other sampling methods**:
> > Thank you for your question regarding the evaluation of the RAG-based approach in comparison to other sampling methods. We chose Retrieval-Augmented Generation (RAG) due to the challenges associated with obtaining detailed and semantic-level information from ECG data, especially when using publicly available datasets. The RAG approach allows us to generate potential waveform details that are not readily available in existing datasets.
> >
> >     To further validate the effectiveness of RAG, we conducted an experiment, now detailed in the newly added Appendix A, where we compared the performance of models trained using RAG-generated data against models trained using data generated directly by a large language model (LLM) without the retrieval component. Our findings indicate that the RAG-based approach outperforms the direct LLM-based method, particularly in generating more accurate and contextually relevant waveform descriptions.
> >
> >     We acknowledge the value of your suggestion to explore other methods for creating pretraining datasets, and we recognize the potential benefits of comparing RAG with alternative approaches. While our current study focuses on the advantages of RAG, we will investigate and benchmark other sampling methods in future work to further improve the quality of ECG data analysis and model performance.
> >
> > 11. **BioLinkBERT for text encoding**:
> > Thank you for your question regarding the use of BioLinkBERT for text encoding. We have added a justification in Section 3.2 of the manuscript to clarify this choice.
> >
> >     BioLinkBERT is an extension of the standard BERT model, specifically designed to improve the understanding of biomedical texts. Unlike traditional BERT, which processes each document independently, BioLinkBERT is pretrained on biomedical literature from PubMed. This pretraining takes advantage of natural links between documents, such as citations and references. This makes BioLinkBERT particularly well-suited for tasks that require a deep understanding of biomedical concepts and terminology.

---

### Decision · Action_Editor_YrY2 · 2024-09-14

**Recommendation:** Accept as is

**Comment:**

The reviewers generally agree that this is solid empirical work with clear writing and convincing results. The major drawback is the lack of more novel methodology; however, for a contribution in a specialized domain like ECG, I think this is okay. Other reviewers' comments are addressed in the rebuttal.

**Audience:**

This paper pre-trains a foundation model for a relatively underexplored modality. Beyond practitioners in this area, readers of TMLR who are interested in applying such methods to their own domains will find this interesting.

**Claims And Evidence:**

This paper claims to construct a large-scale dataset of ECG-text pairs for pre-training the "ECG Semantic Integrator" model. The paper backs up these claims with convincing experiments. Ablations in Appendix A, including new ablations added during the rebuttal, analyze the impact of design decisions of the RAG pipeline on corpus construction. Results on existing test datasets show the abilities of the model in fine-tuning, probing, and zero-shot settings.